# WKV-sharing embraced random shuffle RWKV high-order modeling for pan-sharpening

**Man Zhou, Xuanhua He[1], Danfeng Hong[2], Bo Huang[3]***

[1] University of Science and Technology of China
[2] Southeast University
[3] The University of Hong Kong

## Abstract

Pan-sharpening aims to generate a spatially and spectrally enriched multi-spectral image by integrating information from low-resolution multi-spectral image and texture-rich panchromatic counterpart. In this work, we propose a WKV-sharing embraced random shuffle RWKV high-order modeling paradigm for pan-sharpening from Bayesian perspective, coupled with random weight manifold distribution training strategy derived from Functional theory to regularize the solution space adhering to the following principles: **1) Random-shuffle RWKV.** Recently, the Vision RWKV model, with its inherent linear complexity in global modeling, has inspired us to explore its untapped potential in pan-sharpening tasks. However, its attention mechanism, relying on a recurrent bidirectional scanning strategy, suffers from biased effects and demands significant processing time. To address this, we propose a novel Bayesian-inspired scanning strategy called Random Shuffle, complemented by a theoretically-sound inverse shuffle to preserve information coordination invariance, effectively eliminating biases associated with fixed sequence scanning. The Random Shuffle approach mitigates preconceptions in global 2D dependencies in mathematical expectation, providing the model with an unbiased prior. In line with similar spirit of Dropout, we introduce a testing methodology based on Monte Carlo averaging to ensure the model's output aligns more closely with expected results. **2) WKV-sharing high-order.** Regarding KV's attention score calculation in spatial mixer of RWKV, we leverage WKV sharing mechanism to transfer WKV activations across RWKV layers, achieving lower latency and improved trainability, and revisit the channel mixer in RWKV, originally a first-order weighting function, and redevelop its high-order potential by sharing the gate mechanism across RWKV layer. Comprehensive experiments across pan-sharpening benchmarks demonstrate our model's effectiveness, consistently outperforming state-of-the-art alternatives.

## 1 Introduction

**RWKV modeling.** Transformer-based methods have surpassed traditional CNNs in pan-sharpening performance, yet their global attention mechanisms-$\text{softmax}(\mathbf{q}_i, \mathbf{k}_j), \forall i, j \in \{\mathbf{mn} \times \mathbf{mn}\}$ incur quadratic computational complexity $O((\mathbf{mn})^2)$, rendering them impractical for large-scale applications in Fig. 1. The Vision RWKV model, with its linear complexity in global modeling, has recently emerged as a promising alternative for pan-sharpening.

$$\texttt{Spatial-mixer:} wkv = \texttt{Re-WKV}(\mathbf{K}_s, \mathbf{V}_s), \tag{1}$$

$$\mathbf{O}_s = \texttt{Mapping}(\sigma(\mathbf{R}_s) \odot wkv), \tag{2}$$

---

*corresponding author

39th Conference on Neural Information Processing Systems (NeurIPS 2025).

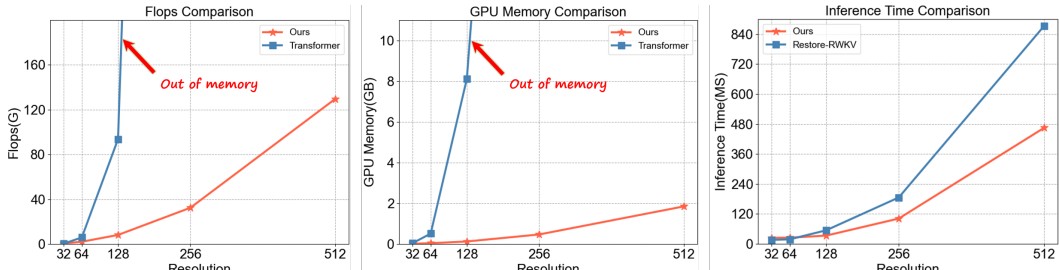

Figure 1: Comparison of Memory, FLOPs, and Inference Time across different scales for Transformer, Restore-RWKV, and our proposed Random-Shuffle RWKV. Unlike traditional transformers, our proposed Random-Shuffle RWKV significantly reduces both memory usage and FLOPs, especially at larger scales where transformers encounter out-of-memory issues. In terms of inference time, our design outperforms the standard RWKV architecture, achieving several-fold reductions in runtime.

$$\texttt{Channel-mixer:} \mathbf{X}_c = \texttt{Omni-Shift}(\mathrm{LN}(\mathbf{O}_s)), \tag{3}$$

$$\mathbf{R}_c, \mathbf{V}_c = \texttt{Mapping}, \gamma(\texttt{Mapping})(\mathbf{X}_c), \tag{4}$$

$$\mathbf{O}_c = \texttt{Mapping}((\sigma(\mathbf{R}_c) \odot \mathbf{V}_c)), \tag{5}$$

where $\texttt{Mapping}(.)$ denotes the non-lineally equipped multi-layer perception, $\sigma$ and $\gamma$ indicate sigmoid and squared ReLU activations respectively. Nonetheless, its attention mechanism, which employs a recurrent bidirectional scanning strategy $\texttt{Re-WKV}(.)$, suffers from biased effects and requires substantial processing time, underscoring the need for further refinement to fully leverage its potential in multi-modal image fusion domain, as indicated in Fig. 2.

**Random shuffle.** To address this, we propose a novel Bayesian-inspired Random Shuffle scanning strategy, complemented by a theoretically-sound inverse shuffle to preserve information coordination invariance, effectively eliminating biases associated with fixed sequence scanning.

$$\texttt{Spatial-mixer:} (\mathbf{K}_s, \mathbf{V}_s) = \texttt{RS}(\mathbf{K}_s, \mathbf{V}_s), \tag{6}$$

$$wkv = \texttt{WKV}(\mathbf{K}_s, \mathbf{V}_s), \tag{7}$$

$$\mathbf{O}_s = \texttt{Mapping}(\sigma(\mathbf{R}_s) \odot wkv), \tag{8}$$

$$\mathbf{O}_s = \texttt{IS}(\mathbf{O}_s) \tag{9}$$

where $\texttt{RS}(.)(.)$ is the random shuffle function and $\texttt{IS}(.)$ is the corresponding inverse shuffle function. The Random Shuffle approach mitigates preconceptions in global 2D dependencies in mathematical expectation, providing the model with an unbiased prior. In line with similar spirit of Dropout, we introduce a testing methodology based on Monte Carlo averaging to ensure the model's output aligns more closely with expected results in Fig. 6.

$$\texttt{Expectation}: \mathbf{O}_s = \mathbb{E}_S[\texttt{RS}, \texttt{WKV}], \tag{10}$$

$$\texttt{Monte-Carlo estimation:} \mathbf{O}_s \approx \frac{1}{\mathrm{M}} \sum_{i=1}^{\mathrm{M}} [\texttt{RS}, \texttt{WKV}] \tag{11}$$

**High-order.** Despite the remarkable progress, existing methods primarily employ the spatial cross-attention and channel-wise scaling mechanisms, which only exploit second-order properties in a cascaded manner, thereby limiting higher-order interaction capabilities. Furthermore, the cascaded second-order interaction paradigm only captures multiple second-order interactions and struggles to balance commendable performance with resource-intensive computations, posing challenges for practical applications, as illustrated in Fig. 1. To address these challenges, our investigation reveals that attention fundamentally operates as a first-order linear weight function

$$\mathbf{O}_j = \mathrm{sigmoid}(\mathbf{R}_c) \cdot \mathbf{V}_c, \tag{12}$$

$$\mathbf{O}_s = \mathrm{sigmoid}(\mathbf{R}_s) \odot wkv, \tag{13}$$

$$0 < \mathrm{sigmoid}(\mathbf{R}_c(i)) < 1, \quad \sum_i \mathrm{sigmoid}(\mathbf{R}_c(i)) = 1, \quad \forall i \tag{14}$$

Mathematically, for any function $p(x)$ satisfying two constraints of $0 \le p(x) \le 1$, $\sum_x p(x) = 1$ and acting as first-order statistic calculating, it equals to

$$p(\mathbf{R}_c) \propto \mathrm{sigmoid}(\mathbf{R}_c(i)), \tag{15}$$

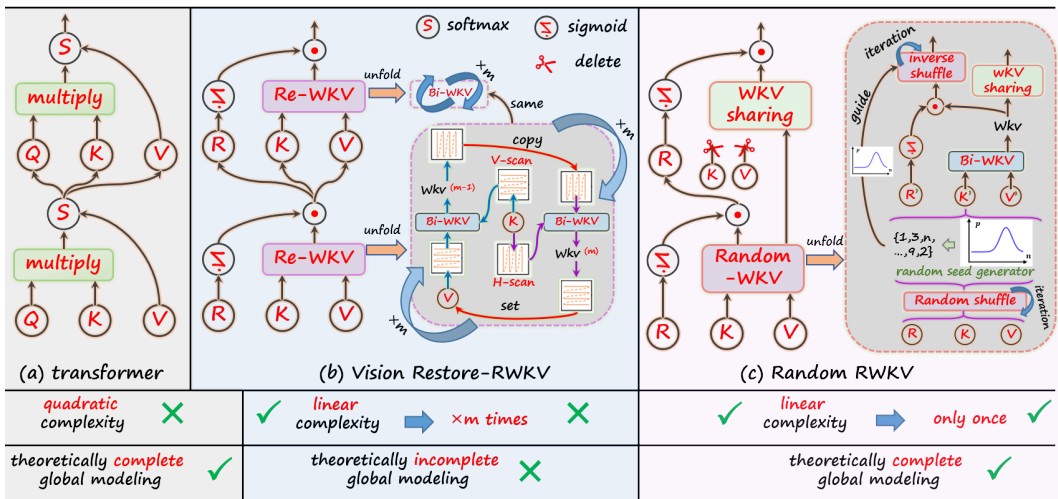

Figure 2: Comparison between previous transformer, Vision Re-RWKV and our proposed random shuffle high-order RWKV paradigm. The typical (a) transformer architecture suffers from quadratic complexity, often resulting in out-of-memory errors when processing high-resolution scenes. Additionally, (b) Vision Re-RWKV performs horizontal and vertical interactions over W and K through m iterations, which, from a Bayesian perspective, can lead to biased effects and incomplete global modeling. The recursive computation process further introduces increased latency. In contrast, (c) our proposed Random RWKV retains the benefits of theoretically incomplete global modeling but operates with linear complexity, offering a more globally effective receptive field. This approach eliminates the issues of memory overload and latency while ensuring efficient modeling.

$$\mathbf{O}_j = \int_0^1 p(\mathbf{R}_c)\mathbf{V}_c d\mathbf{v} \approx \mathbf{E}(\mathbf{v}_c), \tag{16}$$

where $\mathbf{E}(\cdot)$ signifies first-order expectation calculating. This insight enables us to replace the conventional cascaded second-order interaction sequence with efficient high-order modeling through tailored attention sharing.

$$\texttt{Gate potential:} g^{(i-1)} = \text{sigmoid}(\mathbf{R}_c(i)), \tag{17}$$

$$g^{(i)} \leftarrow g^{(i-1)}. \tag{18}$$

Regarding KV's attention score calculation in spatial mixer, we leverage WKV-sharing mechanism to transfer KV activations across RWKV layers, achieving lower latency and improved trainability, and revisit the channel mixer in RWKV, originally a first-order weighting function, and redevelop its high-order potential by sharing the gate mechanism across RWKV layer.

$$\texttt{WKV sharing:} wkv^{(i-1)} = \texttt{WKV}(\mathbf{K}_s, \mathbf{V}_s), \tag{19}$$

$$wkv^{(i)} \leftarrow wkv^{(i-1)}. \tag{20}$$

**Solutions.** In this work, we introduce a WKV-sharing embraced random shuffle RWKV high-order modeling paradigm for pan-sharpening, integrating a Bayesian-inspired Random Shuffle scanning strategy to eliminate biases associated with traditional fixed-sequence scanning in Fig. 3. This approach is complemented by a WKV-sharing mechanism that transfers KV activations across RWKV layers, enhancing trainability and reducing latency while unlocking high-order potential in the channel mixer. Additionally, we implement a random weight, based on Functional theory, to effectively regularize the optimization space, surpassing the limitations of traditional fixed-point loss functions. Extensive experiments across pan-sharpening benchmarks—demonstrate that our model, by harnessing high-order RWKV modeling, significantly enhances the ability to exploit multi-modal synergies, leading to superior performance compared to state-of-the-art methods.

## 2 Proposed Method

In this section, we begin by reviewing the overview of the proposed pan-sharpening network, as depicted in Fig. 3. We then delve into the core building blocks of our approach, which comprise three

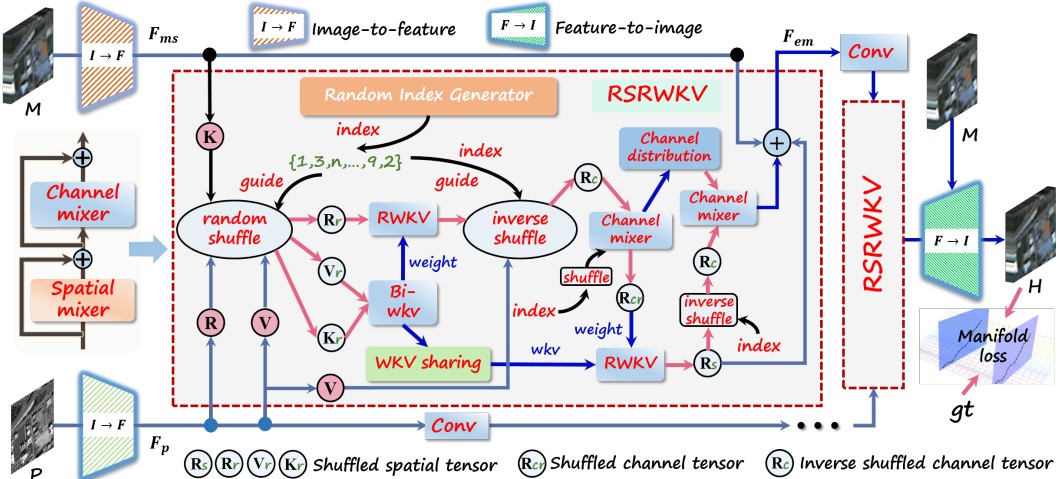

Figure 3: The detailed framework of the proposed Random Shuffle high-order RWKV paradigm (RS-RWKV). RSRWKV treats each feature as a dynamic entity, leveraging a Random Shuffle mechanism, which dynamically alters scanning sequences to enhance global contextual awareness and reduce biases inherent in traditional sequential approaches. By integrating a Bayesian-inspired scanning strategy, the framework effectively addresses the limitations of fixed sequence processing, promoting a more robust understanding of feature relationships. Additionally, the design incorporates a WKV-sharing mechanism that allows for efficient sharing of key-value activations across layers, significantly reducing latency while improving the model's ability to capture intricate inter-dependencies. This synergistic design not only optimizes computational efficiency but also enriches feature representation.

critical components: (a) random shuffle scanning strategy within RWKV's spatial mixer, coupled with Monte-Carlo expectation estimation during inference, (b) high-order potential of gate mechanism within RWKV's channel mixer, and (c) WKV-sharing embraced spatial mixer modeling.

## 2.1 Overview Framework

**Structure Flow.** Given an PAN image, $\mathbf{I}_{\mathcal{P}} \in \mathbb{R}^{\mathrm{H} \times \mathrm{W} \times 1}$, and a low-resolution multi-spectral image, $\mathbf{I}_{\mathcal{M}} \in \mathbb{R}^{\mathrm{h} \times \mathrm{w} \times \mathrm{C}}$, we adopt the separate dual-branch modality-aware encoders to project $\mathbf{I}_{\mathcal{P}}$ and the up-sampled $\mathbf{I}_{\mathcal{M}}$, yielding $\mathbf{F}_{\mathcal{P}} \in \mathbb{R}^{\mathrm{H} \times \mathrm{W} \times \mathrm{C}}$ and $\mathbf{F}_{\mathcal{M}} \in \mathbb{R}^{\mathrm{H} \times \mathrm{W} \times \mathrm{C}}$. Subsequently, the extracted modality-aware shallow-level features are passed through the proposed core RWKV high-order paradigm in a sequential manner as

$$\mathbf{F}_{\mathcal{P}}, \mathbf{F}_{\mathcal{M}} = \mathrm{E}_{\mathcal{P}}(\mathbf{I}_{\mathcal{P}}), \mathrm{E}_{\mathcal{M}}(\mathbf{I}_{\mathcal{M}}) \tag{21}$$

Where $\mathrm{E}_{\mathcal{P}}(\cdot)$ and $\mathrm{E}_{\mathcal{M}}(\cdot)$ signify the feature extraction encoders for the PAN and multi-spectral modalities, respectively. Then, we employ the successively designed KV-cache embraced random shuffle RWKV high-order modeling, yielding across modality-aware features $\mathbf{F}_{\mathcal{P}}$ and $\mathbf{F}_{\mathcal{M}}$

$$\mathbf{F}_{\mathcal{M}}^{(1)}, \mathbf{F}_{\mathcal{P}}^{(1)} = \mathrm{RSRWKV}_{(i-1)}(\mathbf{F}_{\mathcal{P}}, \mathbf{F}_{\mathcal{M}}), \tag{22}$$

$$\mathbf{F}_{\mathcal{M}}^{(i)}, \mathbf{F}_{\mathcal{P}}^{(i)} = \mathrm{RSRWKV}_{(i)}(\mathbf{F}_{\mathcal{M}}^{(i-1)}, \mathbf{F}_{\mathcal{P}}^{(i-1)}), \ \ i \in \{1, L\} \tag{23}$$

where $L$ indicates the iteration number of our RSRWKV. Finally, the transformed deep-level features are projected back into the image space to generate the fused result, $I_{\mathcal{F}} \in \mathbb{R}^{\mathrm{H} \times \mathrm{W} \times \mathrm{C}}$ from the encoder in conjunction with the $1 \times 1$ convolution unit as

$$\mathbf{I}_{\mathcal{F}} = \mathrm{D}_{\mathcal{C}}(\mathbf{F}_{\mathcal{H}}) + \mathrm{Up}(\mathbf{I}_{\mathcal{M}}) \tag{24}$$

where $\mathrm{Up}(.)$ and $\mathrm{D}_{\mathcal{C}}(\cdot)$ represent the up-sampling and the corresponding decoder, respectively.

**Supervision Flow.** In this study, we introduce a novel loss function for optimizing the pan-sharpening process and enhancing results, independent of the structure design. Our proposed loss function comprises two components: spatial domain loss $\mathcal{L}_s$ and implicit frequency-decomposition manifold loss $\mathcal{L}_m$, as illustrated in Fig. 5. Prior pan-sharpening methods typically employ pixel losses with local guides in the spatial domain. However, our approach incorporates an additional frequency-decomposition manifold loss, utilizing random weight derived Taylor's unfolding manifold to regularize the optimization space, resulting in improved pan-sharpening performance.

$$\mathtt{Structure\ loss:} \mathcal{L}_s = \mathtt{L1}(\mathbf{I}_{\mathcal{F}}, \mathbf{I}_{\mathcal{H}}), \tag{25}$$

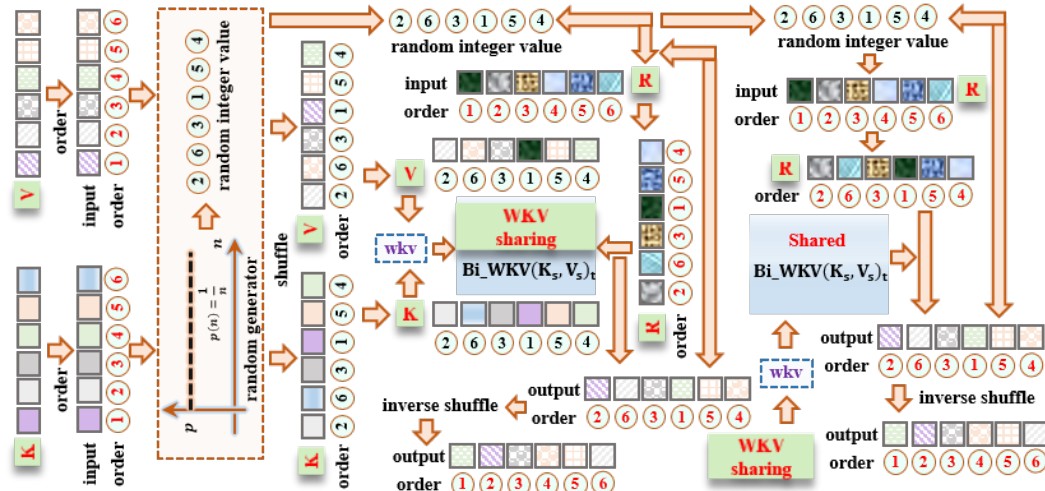

Figure 4: The detailed flowchart of RSRWKV modeling. The square box sequences, denoted as R, K, and V, represent the three components within the RWKV framework and are linked to the circular numbers generated by the random distribution generator, which indicate the random shuffle guidance as `order`. This guidance facilitates the implementation of the Bayesian-inspired scanning strategy and the theoretically sound inverse shuffle.

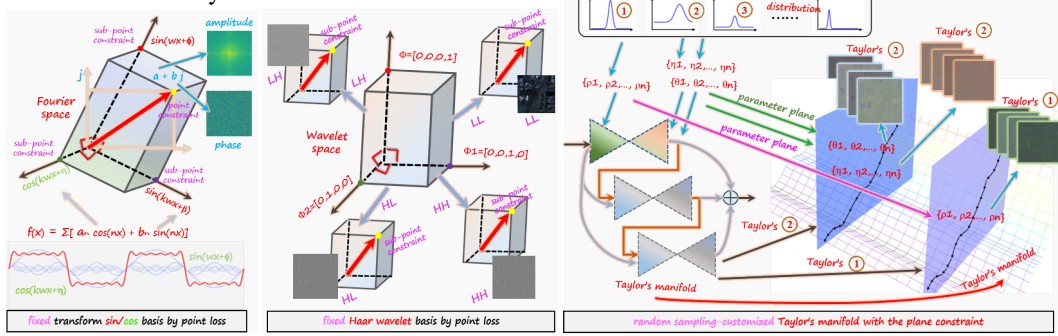

Figure 5: Comparison between Point Loss and Our Customized Manifold Loss. Traditional frequency point losses, such as those based on Fourier and wavelet transforms, aim to constrain the reconstructed output to possess richer textures.

$$\texttt{Manifold loss:} \mathcal{L}_m = \texttt{Taylor's}(\mathbf{I}_\mathcal{F}, \mathbf{I}_\mathcal{H}; \theta_e), \tag{26}$$

$$\theta_e \sim \{\texttt{Xavier, Kaiming init, Gaussian}(0,1)\} \tag{27}$$

where $\mathbf{I}_\mathcal{H}$ represent the ground truth, L1 indicates the 1-norm, and $\theta_e$ denotes the random weights for each epoch within Taylor's unfolding manifold plane. The total loss function is remarked as

$$\mathcal{L} = \mathcal{L}_s + \lambda \mathcal{L}_m \tag{28}$$

## 2.2 Random shuffle RWKV

**Preliminaries of RWKV.** Vision RWKV consists of a spatial-mixer module and a channel-mixer module. Given the feature $F_\mathcal{M}^{(i)}$, $F_\mathcal{P}^{(i)}$ and flattened to a one-dimensional sequence $\mathbf{X}_\mathcal{M} \in \mathbb{R}^{\mathrm{T} \times \mathrm{C}}$ and $\mathbf{X}_\mathcal{P} \in \mathbb{R}^{\mathrm{T} \times \mathrm{C}}$, where $\mathrm{T} = \mathrm{H} \times \mathrm{W}$ represents the number of tokens, $\mathbf{X}_\mathcal{P}$ and $\mathbf{X}_\mathcal{M}$ are initially processed by a layer normalization operation, followed by an Token shift layer

$$\texttt{Spatial-mixer:} wkv = \texttt{Re-WKV}(\mathbf{K}_s, \mathbf{V}_s), \tag{29}$$

$$\mathbf{O}_s = \texttt{Mapping}(\sigma(\mathbf{R}_s) \odot wkv), \tag{30}$$

Subsequently, $\mathbf{X}_\mathcal{M}$ and $\mathbf{X}_\mathcal{P}$ are processed by three parallel projected linear modules, yielding outputs receptance $\mathbf{R_s}$, key $\mathbf{K_s}$, and value $\mathbf{V_s}$:

$$\mathbf{R}_s = \mathbf{X}_\mathcal{M}\mathbf{W}_\mathrm{R}, \quad \mathbf{K}_s = \mathbf{X}_\mathcal{P}\mathbf{W}_\mathrm{K}, \quad \mathbf{V}_s = \mathbf{X}_\mathcal{P}\mathbf{W}_\mathrm{V} \tag{31}$$

where $\mathbf{W}_\mathrm{R}$, $\mathbf{W}_\mathrm{K}$ and $\mathbf{W}_\mathrm{V}$ denote the linear layer. $\mathbf{K}_s$ and $\mathbf{V}_s$ serve as inputs to the `Re-WKV` attention mechanism. Nonetheless, its attention mechanism, which employs a recurrent bidirectional scanning

strategy `Re-WKV(.)`. To enhance the global receptive field of the WKV attention, the Bi-WKV layer is applied iteratively **Q** times, as detailed as following:

$$wkv_t = \texttt{Re-WKV}(\mathbf{K}_s, \mathbf{V}_s)_{\mathbf{t}} = \frac{\sum_{i=0,i\neq t}^{T-1} e^{-(|t-i|-1)/T\cdot w + \mathbf{k}_i}\mathbf{v}_i + e^{u+\mathbf{k}_t}\mathbf{v}_t}{\sum_{i=1,i\neq t}^{T} e^{-(|t-i|-1)/T\cdot w + \mathbf{k}_i} + e^{u+\mathbf{k}_t}}, \tag{32}$$

$$\mathbf{wkv} = \texttt{Re-WKV}_{(\mathbf{Q})}(\mathbf{K}_s, \mathbf{V}_s). \tag{33}$$

Here, $wkv_t$ denotes the attention for the t-th token, $u$ and $w$ serve as hyperparameters within the attention mechanism. The $\mathbf{k}_i$ and $\mathbf{v}_i$ represent the i-th spatial tokens derived from $\mathbf{K}_s$ and $\mathbf{V}_s$, respectively. The resulting $wkv$ is then passed through a Sigmoid function and subsequently multiplied by $\mathbf{R}_s$. This product is added to $F_{\mathcal{M}}^{(i)}$ to obtain the final spatially mixed output:

$$\mathbf{O_{sf}} = \mathbf{O_s} + F_{\mathcal{M}}^{(i)}. \tag{34}$$

Followed, the enriched MS feature $\mathbf{O_{sf}}$ from spatial-mixer's output concatenated with PAN feature $F_{\mathcal{P}}^{(i)}$ is feed into channel-mixer as

$$\texttt{Channel-mixer:} \mathbf{O}_{sc} = \text{Concate}(\mathbf{O_{sf}}, F_{\mathcal{P}}^{(i)}), \tag{35}$$
$$\mathbf{X}_c = \texttt{Omni-Shift}(\text{LN}(\mathbf{O}_{sc})), \tag{36}$$
$$\mathbf{R}_c, \mathbf{V}_c = \texttt{Mapping}, \gamma(\texttt{Mapping})(\mathbf{O}_{sc}), \tag{37}$$
$$\mathbf{O}_c = \texttt{Mapping}((\sigma(\mathbf{R}_c) \odot \mathbf{V}_c)), \tag{38}$$

Nonetheless, its attention mechanism, which employs a recurrent bidirectional scanning strategy `Re-WKV(.)`, suffers from biased effects and requires substantial processing time.

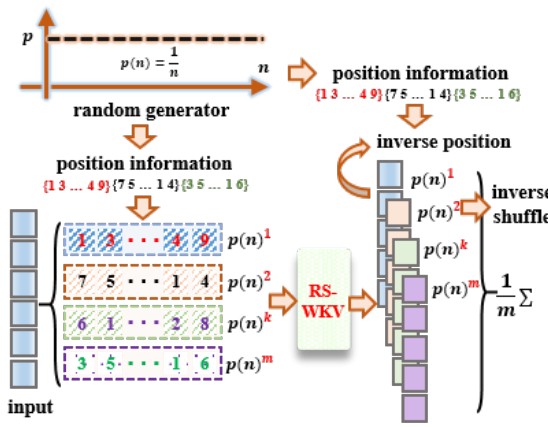

Figure 6: Testing RS-WKV with Monte Carlo Averaging. The random generator operates on a uniform distribution to produce integer values that serve as position information within the shuffling strategy. This position information is used to shuffle the input data accordingly. The shuffled input is then processed by RS-WKV to capture long-range cross-modality dependencies. To maintain information invariance, we apply an inverse shuffle using the cached position information to obtain the weighted output.

**Spatial mixer.** To address this, we propose a novel Bayesian-inspired scanning strategy called Random Shuffle, complemented by a theoretically-sound inverse shuffle to preserve information coordination invariance, effectively eliminating biases associated with fixed sequence scanning.

$$\texttt{Spatial-mixer:}(\mathbf{K}_s, \mathbf{V}_s) = \texttt{RS}(\mathbf{K}_s, \mathbf{V}_s), \tag{39}$$
$$wkv = \texttt{WKV}(\mathbf{K}_s, \mathbf{V}_s), \tag{40}$$
$$\mathbf{O}_s = \texttt{Mapping}(\sigma(\mathbf{R}_s) \odot wkv), \tag{41}$$
$$\mathbf{O}_s = \texttt{IS}(\mathbf{O}_s) \tag{42}$$

The Random Shuffle approach mitigates preconceptions in global 2D dependencies in mathematical expectation, providing model with an unbiased prior in Fig. 4. In line with similar spirit of Dropout, we introduce a testing methodology based on Monte Carlo averaging to ensure the model's output aligns more closely with expected results through layered expectations. Therefore, the computation of the random shuffle during testing can be expressed as follows, where

$$\texttt{Expectation:} \mathbf{O}_s = \mathbb{E}_S[\texttt{RS}, \texttt{WKV}], \tag{43}$$

$$\texttt{Monte-Carlo estimation:} \mathbf{O}_s \approx \frac{1}{\text{M}} \sum_{i=1}^{\text{M}} [\texttt{RS}, \texttt{WKV}] \tag{44}$$

It seems that the testing time will be scaled by M, which is the number of averaged forward passes. However, the multiple forward passes can be conducted concurrently with modern accelerators, which significantly reduces the testing time in Fig. 2. Specifically, this acceleration can be done by transferring an input to GPU(s) and setting a mini-batch comprising the same input multiple

times. `WKV` shuffles independently along the batch dimension. After one forward pass through `WKV`, averaging over the mini-batch yields the Monte-Carlo estimation.

**High-order channel mixer.** However, existing methods primarily employ the spatial cross-attention and channel-wise scaling mechanism, which only exploits second-order properties in a cascaded manner, thereby limiting higher-order interaction capabilities. Furthermore, the cascaded second-order interaction paradigm only captures multiple second-order interactions and struggles to balance commendable performance with resource-intensive computations. To address this, our investigation reveals that attention fundamentally operates as a first-order linear weight function

$$\mathbf{O}_j = \text{sigmoid}(\mathbf{R}_c) \cdot \mathbf{V}_c, \tag{45}$$

$$\mathbf{O}_s = \text{sigmoid}(\mathbf{R}_s) \odot wkv, \tag{46}$$

$$0 < \text{sigmoid}(\mathbf{R}_c(i)) < 1, \quad \sum_i \text{sigmoid}(\mathbf{R}_c(i)) = 1, \quad \forall i \tag{47}$$

In mathematically, for any function $p(x)$ satisfying two constraints of $0 \leq p(x) \leq 1, \quad \sum_x p(x) = 1$ and acting as first-order statistic calculating, it equals to

$$p(\mathbf{R}_c) \propto \text{sigmoid}(\mathbf{R}_c(i)), \tag{48}$$

$$\mathbf{O}_j = \int_0^1 p(\mathbf{R}_c)\mathbf{V}_c d\mathbf{v} \approx \mathbf{E}(\mathbf{v}_c), \tag{49}$$

This insight enables us to replace the conventional cascaded second-order interaction sequence with efficient high-order modeling through tailored attention sharing.

$$\texttt{Gate potential:} g^{(i-1)} = \text{sigmoid}(\mathbf{R}_c(i)), \tag{50}$$

$$g^{(i)} \leftarrow g^{(i-1)}. \tag{51}$$

In the context of first-order statistical expectation of a variance tensor, referring to the definition, we assume any probability distribution $p(\mathbf{v})$ that satisfies two constraints: $0 \leq p(\mathbf{v}) \leq 1, \quad \sum_i p(\mathbf{v}) = 1$. Given this, the expectation of $\mathbf{v}_j$ can be expressed as

$$\mathbf{E}(\mathbf{v}_j) = \int_0^1 p(\mathbf{v})\mathbf{v}_j d\mathbf{v} \tag{52}$$

Referring to the definition above, we consider the $\text{sigmoid}(.)$ function, which satisfies the constraints, as a special case of a probability sampling distribution. This allows us to deduce that our investigation reveals attention fundamentally operates as a first-order linear weight function and can be constituted by the simple 1-dimension convolution $\text{Conv}_1$. By leveraging the matrix associative property,

$$g^{(i)} = \text{Conv}_1(g^{(i-1)}), \quad \mathbf{O}_j = \mathbf{v}_c \cdot g^{(i)} \tag{53}$$

$$\texttt{Gate potential:} \mathbf{O}_j = \mathbf{v}_c \cdot \text{Conv}_1(g^{(i-1)}), \tag{54}$$

$$g^{(i)} \leftarrow g^{(i-1)}, \quad \mathbf{O}_j = \text{Conv}_1(\mathbf{v}_c \cdot g^{(i)}) \tag{55}$$

Mathematically equivalent transformation is capable of further mitigating attention collapse.

Similar to high-order channel interactions, we extend the `wkv` calculating within spatial mixer to achieve high-order spatial interactions:

$$\mathbf{O}_s = \text{sigmoid}(\mathbf{R}_s) \odot wkv, \tag{56}$$

the above process signifies first-order expectation calculating.

**WKV-sharing RWKV high-order modeling.** Regarding WKV's attention score calculation in spatial mixer, we leverage WKV-sharing mechanism to transfer WKV activations across RWKV layers and revisit the channel mixer in RWKV, originally a first-order weighting function, and redevelop its high-order potential by sharing the gate mechanism across RWKV layer.

$$\texttt{WKV sharing:} wkv^{(i-1)} = \texttt{WKV}(\mathbf{K}_s, \mathbf{V}_s), \tag{57}$$

$$wkv^{(i)} \leftarrow wkv^{(i-1)}. \tag{58}$$

To facilitate cross-order information integration, we enhance the representation of cross-modality interactions by incorporating diverse information in a cross-order manner. This enhancement enables the generation of more informative representations by leveraging the observation that different orders tend to capture diverse and complementary patterns.

$$\mathbf{V}_b \leftarrow \texttt{Conv}(\text{concat}[\mathbf{V}_1, \mathbf{V}_2, \ldots, \mathbf{V}_n]). \tag{59}$$

where $\mathbf{V}_n$ denotes the `n-th` order spatial and channel-wise information within the tailored high-order.

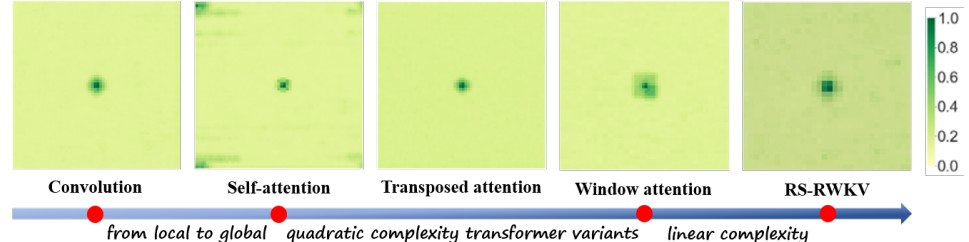

Figure 7: The Effective Receptive Field (ERF) visualization for various models. Our proposed RS-RWKV achieves the most extensive global ERF, demonstrating its superior capacity.

Table 1: Comparison on the WordView-II, WordView-III and GaoFen2 datasets.

| Method | WordView-II | | | | WordView-III | | | | GaoFen2 | | | |
|---|---|---|---|---|---|---|---|---|---|---|---|---|
| | PSNR ↑ | SSIM ↑ | SAM ↓ | ERGAS ↓ | PSNR ↑ | SSIM ↑ | SAM ↓ | ERGAS ↓ | PSNR ↑ | SSIM ↑ | SAM ↓ | ERGAS ↓ |
| SFIM | 34.1297 | 0.8975 | 0.0439 | 2.3449 | 21.8212 | 0.5457 | 0.1208 | 8.9730 | 36.9060 | 0.8882 | 0.0318 | 1.7398 |
| GS | 35.6376 | 0.9176 | 0.0423 | 1.8774 | 22.5608 | 0.5470 | 0.1217 | 8.2433 | 37.2260 | 0.9034 | 0.0309 | 1.6736 |
| Brovey | 35.8646 | 0.9216 | 0.0403 | 1.8238 | 22.5060 | 0.5466 | 0.1159 | 8.2331 | 37.7974 | 0.9026 | 0.0218 | 1.3720 |
| IHS | 35.2962 | 0.9027 | 0.0461 | 2.0278 | 22.5579 | 0.5354 | 0.1266 | 8.3616 | 38.1754 | 0.9100 | 0.0243 | 1.5336 |
| GFPCA | 34.558 | 0.9038 | 0.0488 | 2.1400 | 22.3400 | 0.4826 | 0.1294 | 8.3964 | 37.9443 | 0.9204 | 0.0314 | 1.5604 |
| PNN | 40.755 | 0.9624 | 0.0259 | 1.0646 | 29.9418 | 0.9121 | 0.0824 | 3.3206 | 43.1208 | 0.9704 | 0.0172 | 0.8528 |
| PANNet | 40.8176 | 0.9626 | 0.0257 | 1.0557 | 29.6840 | 0.9072 | 0.0851 | 3.4263 | 43.0659 | 0.9685 | 0.0178 | 0.8577 |
| MSDCNN | 41.3355 | 0.9664 | 0.0242 | 0.9940 | 30.3038 | 0.9184 | 0.0782 | 3.1884 | 45.6874 | 0.9827 | 0.0135 | 0.6389 |
| SRPPNN | 41.4538 | 0.9679 | 0.0233 | 0.9899 | 30.4346 | 0.9202 | 0.0770 | 3.1553 | 47.1998 | 0.9877 | 0.0106 | 0.5586 |
| GPPNN | 41.1622 | 0.9684 | 0.0244 | 1.0315 | 30.1785 | 0.9175 | 0.0776 | 3.2593 | 44.2145 | 0.9815 | 0.0137 | 0.7361 |
| INNformer | 41.6903 | 0.9704 | 0.0227 | 0.9514 | 30.5365 | 0.9225 | 0.0747 | 3.0997 | 47.3528 | 0.9893 | 0.0102 | 0.5479 |
| MutNet | 41.6773 | 0.9705 | 0.0224 | 0.9519 | 30.4907 | 0.9223 | 0.0749 | 3.1125 | 47.3042 | 0.9892 | 0.0102 | 0.5481 |
| SFINet | 41.7244 | 0.9725 | 0.0220 | 0.9506 | 30.5971 | 0.9236 | 0.0741 | 3.0798 | 47.4712 | 0.9901 | 0.0102 | 0.5462 |
| PanFlowNet | 41.8548 | 0.9712 | 0.0224 | 0.9335 | 30.4873 | 0.9221 | 0.0751 | 3.1142 | 47.2533 | 0.9884 | 0.0103 | 0.5512 |
| Ours | **42.0945** | 0.9721 | **0.0214** | **0.9081** | **30.9665** | **0.9266** | **0.0726** | **2.9247** | **47.7144** | **0.9896** | **0.0098** | **0.5229** |

# 3 Experiments over pan-sharpening

To evaluate the performance, we conduct comparative analysis against pan-sharpening. The traditional methods included SFIM [1], Brovey [2], GS [3], IHS [4], and GFPCA [5]. Additionally, we include various deep learning-based techniques, such as PNN [6], PANNET [7], MSDCNN [8], SRPPNN [9], GPPNN [10], MutNet [11], INNformer [12], SFINet [13], and PanFlowNet [14].

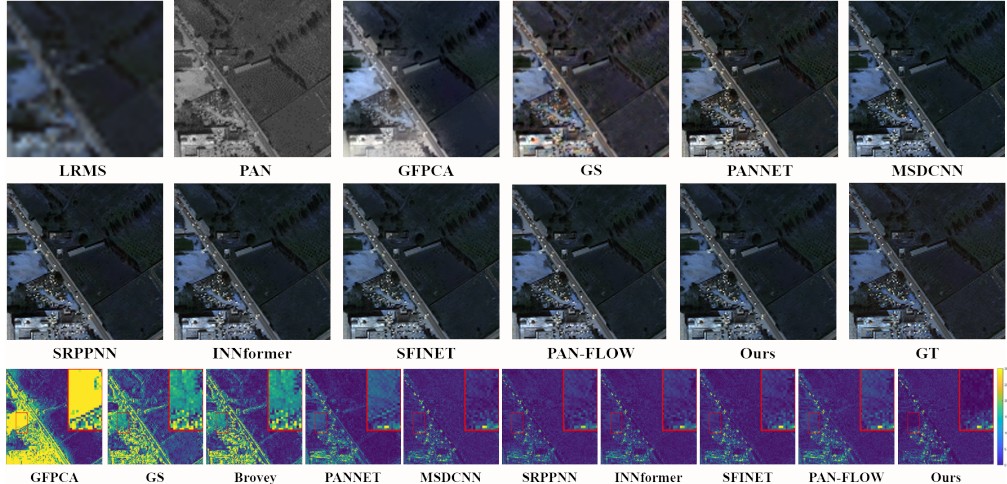

Figure 8: Visual comparisons between other pan-sharpening methods on WorldView-III satellite.

**Comparisons with SOTA.** To assess the performance , we employed a diverse set of metrics, with the results systematically presented in Table 1. These results highlight the outstanding performance of our techniques, clearly demonstrating their superiority over benchmark algorithms across all evaluation criteria. Due to page limit, we present visual comparisons of representative samples from the WorldView-II and WorldView-III datasets in supplementary materials.

**Effect of the Number $L$ of RSRWKV:** To investigate the impact of model size, we conducted ablation studies by varying the number of RSRWKV layers. As illustrated in Table 2, performance

Table 3: Ablation studies on the proposed core designs over the WorldView-II datasets.

| Config | KV-cache | Channel-mixer cache | Random shuffle | Random mainfold loss | PSNR↑ | SSIM↑ | SAM↓ | ERGAS↓ |
|---|---|---|---|---|---|---|---|---|
| (I) | ✗ | ✓ | ✓ | ✓ | 41.9967 | 0.9715 | 0.0222 | 0.9344 |
| (II) | ✓ | ✗ | ✓ | ✓ | 41.9478 | 0.9715 | 0.0221 | 0.9427 |
| (III) | ✓ | ✓ | ✗ | ✓ | 41.9172 | 0.9713 | 0.0224 | 0.9274 |
| (IV) | ✓ | ✓ | ✓ | ✗ | 41.9394 | 0.9713 | 0.0219 | 0.9293 |
| (V) | ✓ | ✓ | ✓ | ✓ | 42.0945 | 0.9721 | 0.0214 | 0.9081 |

improved significantly as the number of RSRWKV components increased, demonstrating a clear benefit from incorporating additional layers. However, it can be observed that this performance enhancement plateaued beyond three components, with only marginal improvements noted upon further increases. To balance performance gains with computational efficiency, we selected $L = 9$ as the default configuration with the model efficacy while maintaining computational load.

**Effect of the core designs:** We conducted a series of ablation studies to systematically investigate the impact of each proposed core designs in Table 3: KV-cache, Channel-mixer cache, Random Shuffle, and Random Manifold Loss. Each experiment involved the removal of one core design from the framework to assess its contribution to overall performance. The results indicate that the absence of any single core design consistently leads to a decline in model performance. Specifically, removing the KV-cache increased latency due to inefficient cross-layer information sharing. Excluding the Channel-mixer cache weakened cross-modal dependency modeling, degrading fusion quality. Removing Random Shuffle introduced fixed-sequence biases, reducing feature diversity. Omitting the Random Manifold Loss destabilized training convergence via unregularized optimization.

Table 2: Comparison on the WorldView-II datasets as the number of RSRWKV increases.

| Number (L) | PSNR↑ | SSIM↑ | SAM↓ | ERGAS↓ |
|---|---|---|---|---|
| tiny (L=3) | 41.9596 | 0.9715 | 0.0220 | 0.9133 |
| small (L=7) | 41.9972 | 0.9716 | 0.0218 | 0.9184 |
| regular (L=9) | 42.0945 | 0.9721 | 0.0214 | 0.9081 |
| Large (L=11) | 42.0976 | 0.9722 | 0.0213 | 0.9076 |

**Effect of ERF:** ERF visualization for various models in Fig. 7. The dark regions in the visualizations represent the extent of the ERF, with a more widespread distribution of darker areas indicating a larger and more effective receptive field. A larger ERF suggests that the model can capture more global context and long-range dependencies.

Among the models compared, our proposed RS-RWKV achieves the most extensive global ERF, demonstrating its superior capacity to integrate information across both local and distant regions,

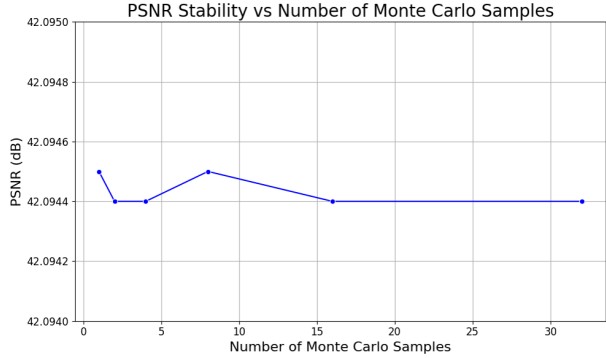

Figure 9: Monte Carlo averaging over random shuffle inference within spatial mixer.

**Effect of Monte Carlo sampling:** Due to the Monte Carlo averaging implemented in our RWKV framework, we explored the effect of varying the sampling number from 1 to 32 on performance, visualized in Fig. 9. Our findings indicated that performance remained remarkably stable across this range, suggesting that utilizing a single sample is sufficient for achieving reliable results. This decision not only minimizes processing time but also simplifies the implementation, making it more efficient for practical applications. Consequently, we adopted a sampling number of one, which allows for quicker model iterations without compromising output quality. This optimization enhances the overall efficiency, making it more suitable for real-world scenarios.

## 4 Conclusion

We propose RS-RWKV, a novel pan-sharpening framework that synergizes multi-modal data through three core innovations: (1) a Random Shuffle strategy to eliminate fixed-scanning bias and enhance global modeling; (2) a KV-cache mechanism for efficient cross-layer activation sharing, reducing latency while improving trainability; and (3) a Random-weight manifold loss to regularize the optimization landscape. Extensive evaluations across pan-sharpening demonstrate our method's superior performance against state-of-the-art baselines.

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

## A   Technical Appendices and Supplementary Material

### A.1   Manifold loss.

Inspired by our previous work, pan-sharpening aims to reconstruct the missing middle and high frequencies. Existing approaches often rely on fixed-point frequency domain loss functions, such as those based on the Discrete Wavelet Transform (DWT) and Discrete Fourier Transform (DFT), which employ fixed orthogonal basis transformations in Fig. 5. These methods can introduce bias into the network's predictions, as the loss functions do not capture the full complexity of the error distribution between network predictions and ground truth from a Bayesian perspective. This complexity makes model optimization challenging and can lead to biased predictions. To address this, we build on Functional Theory to demonstrate that random weight networks, structured within a

strict mathematical manifold, can be formulated as a manifold loss function plane. This formulation effectively regularizes the optimization space, providing an advantage over traditional fixed-point loss functions. Prior research finds the main part and the derivative part of Taylor's Approximations take the same effect as the two competing goals of high-level contextualized information and spatial details of image fusion respectively. Drawing inspiration from image frequency-level decomposition, we leverage Taylor's unfolding manifold, with the weights randomly initialized in each training iteration epoch, to formulate the manifold loss function plane while accounting for the implicit frequency decomposition constraint.

$$\texttt{Taylor's unfolding:} \mathcal{L}_m = \texttt{Taylor's}(\mathbf{I}_F, \mathbf{I}_H; \theta_e), \tag{60}$$

$$\theta_e \sim \{\texttt{Xavier, Kaiming init, Gaussian}(0,1)\} \tag{61}$$

In summary, the contributions of this work are as follows.

- Random Shuffle RWKV for Pan-Sharpening: We propose a novel Random Shuffle scanning strategy within the RWKV framework, inspired by Bayesian principles, to mitigate biases inherent in fixed-sequence scanning. This method enhances global 2D dependency modeling by providing an unbiased prior, improving pan-sharpening performance.

- KV-Cache High-Order Modeling: We introduce a WKV-sharing mechanism to share KV activations across RWKV layers, significantly reducing latency and enhancing trainability. Additionally, we extend the channel mixer in RWKV from a first-order to a high-order function, further boosting the model's capacity to capture complex inter-dependencies.

- Random Weight Manifold Loss: We develop a random weight manifold loss function grounded in Functional theory, which effectively regularizes the optimization space. This approach overcomes the limitations of traditional fixed-point loss functions, leading to better convergence and improved performance in pan-sharpening.

- Extensive Experimental Validation: We conduct comprehensive experiments, demonstrating that our model consistently outperforms state-of-the-art alternatives, establishing a new standard for performance.

## A.2  Limitation & broader impact

A potential limitation is that the proposed framework has not been extensively tested across diverse remote sensing tasks, for example hyperspectral and multi-spectral image fusion. Future studies should validate its generalizability in broader remote sensing application scenarios.

Remote sensing fusion integrates multi-modal data to produce enhanced observations, critical for environmental monitoring (e.g., forest loss), disaster response (wildfire/flood tracking), and sustainable development (agriculture, urban heat analysis). By enabling cost-effective access to precise data, it democratizes global resources—helping developing nations tackle climate and food security challenges—while advancing cross-disciplinary geoscience research.

## A.3  Related work

Traditional pan-sharpening methods are typically classified into three primary categories: Component Substitution (CS), Multi-resolution Analysis (MRA), and Variational Optimization (VO) approaches. CS methods, such as intensity hue-saturation (IHS) fusion, principal component analysis (PCA), Brovey transforms, and Gram-Schmidt (GS) orthogonalization, are widely used [15, 16]. Enhancements to these methods include nonlinear IHS (NIHS) to reduce spectrum distortion and adaptive techniques like the GSA method [17, 18]. Despite their practicality, CS and MRA methods often introduce artifacts into the fused images. VO methods have emerged as alternatives to address spectral distortion and improve the spatial resolution of multi-spectral images. For example, P+XS pan-sharpening posits that the PAN image can be modeled as a linear combination of high-resolution multi-spectral (HRMS) bands, with the upsampled low-resolution multi-spectral (LRMS) image approximating a blurred HRMS image [19]. VO approaches incorporate constraints such as dynamic gradient sparsity (SIRF), local gradient constraints (LGC), and group low-rank constraints for texture similarity (ADMM) [20, 21, 22]. Despite their sophistication, VO methods often require manual parameter tuning and may struggle to capture structural relationships within images, potentially leading to suboptimal performance.

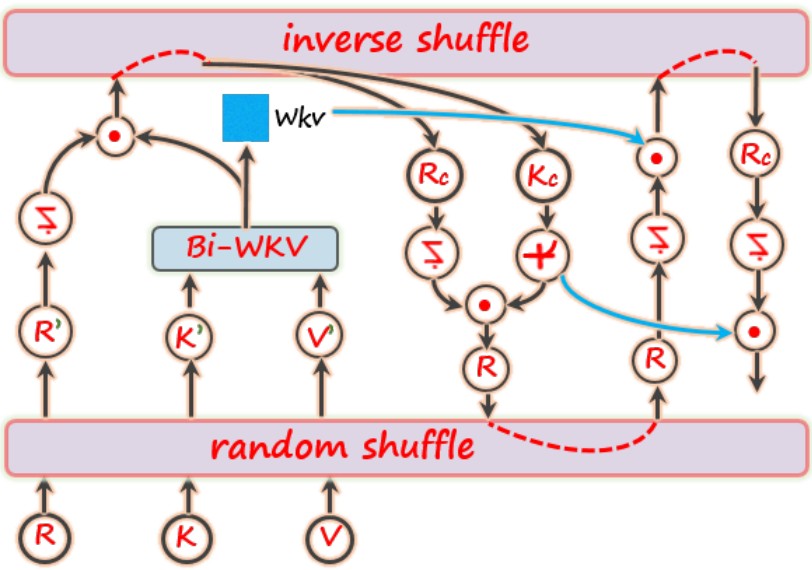

Figure 10: The illustration of high-order WKV sharing and channel mixer cache mechanism at adjacent steps. The calculated values from the previous step within the spatial mixer are shared with the subsequent step, facilitating the implementation of the WKV sharing and replacing the complex computation of `wkv`. Similarly, the channel-wise distribution within the channel mixer employs an analogous high-order sharing mechanism. Each stage adheres to a consistent random shuffle guidance, ensuring cohesive integration across the framework.

Convolutional Neural Networks (CNNs) have transformed computer vision with their prowess in nonlinear fitting and feature extraction, making them crucial for hyperspectral and remote sensing image analysis. Recent advancements in pan-sharpening focus on CNN-based approaches [23, 24]. Masi *et al.* [6] were among the first to apply CNNs to pan-sharpening, demonstrating superior results compared to traditional methods. Yang *et al.* [7] extended this work by incorporating residual blocks [25] into a deeper CNN architecture, while Wu *et al.* [26] introduced a multi-scale module to enhance the CNN structure. Cai *et al.* [9] further improved performance by utilizing multi-scale image inputs within the backbone network. A new category of model-driven CNNs has recently gained traction, integrating physical insights into optimization-based tasks. Xu *et al.* [10] applied distinct priors for PAN and MS images within a structured CNN framework, enhancing interpretability. Xie *et al.* [27] incorporated an optimization algorithm into a CNN architecture, while Tian *et al.* [28] and Wu *et al.* [29] combined VO techniques with deep residual CNNs. Zhou *et al.* [11] introduced a novel pan-sharpening framework driven by mutual information, which enhances information representation through complementary learning between PAN and MS modalities, thereby reducing redundancy and significantly improving pan-sharpening performance.

## A.4 Feature Visualization

To verify the contributions of the proposed random shuffle RWKV high-order modeling mechanism, we analyze the feature maps corresponding to the `input`, the `Random shuffled` features within the RS-RWKV framework, the `Output` from the inverse shuffled component, and the `Enhanced` feature generated by summing the `input` with the `Output`. As detailed in Section 2.2, the randomly shuffled feature `Random shuffled` exhibits a chaotic state, aligning with theoretical expectations. Fig. 12 demonstrates that the `Output` from the inverse shuffled component effectively captures global information while emphasizing cross-modality detail. By integrating the extracted detailed information into the `input`, the final `Enhanced` feature provides a more informative and comprehensive representation of the input image. These findings indicate that the designed random shuffle RWKV high-order modeling mechanism successfully fuses global information from multiple modalities, leading to improved model performance.

Furthermore, we conducted a visualization of the key components within the RWKV framework in Fig. 11, specifically `R`, `K`, and `V`, while systematically varying the stage-wise RS-RWKV from bottom to top. The results indicate that with an increase in stages, a progressively larger number of features

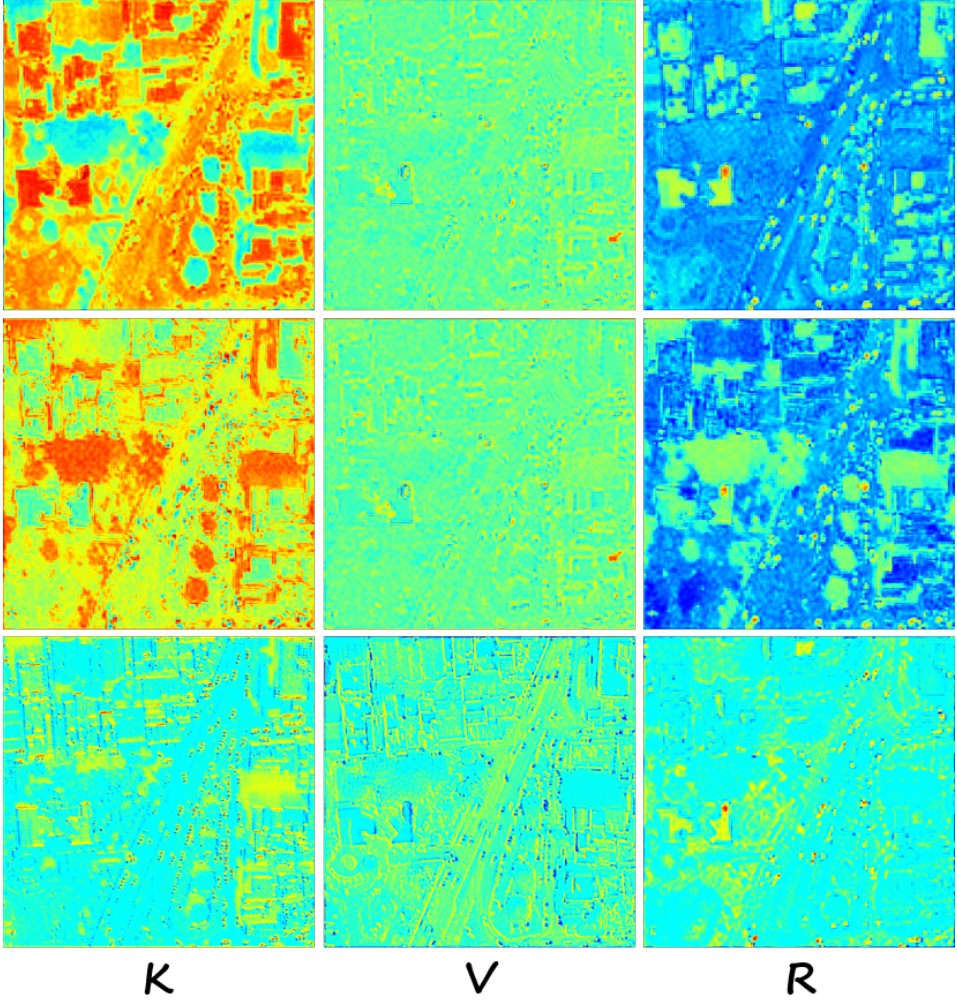

Figure 11: The key fundamental factors of RSRWKV over pan-sharpening by varying stages.

are activated. Notably, the features corresponding to V and K exhibit a complementary relationship, which is advantageous for the extraction of salient features. This phenomenon is consistent with the design principles of the RWKV framework, reinforcing its capacity to effectively harness high-order interactions for improved performance.

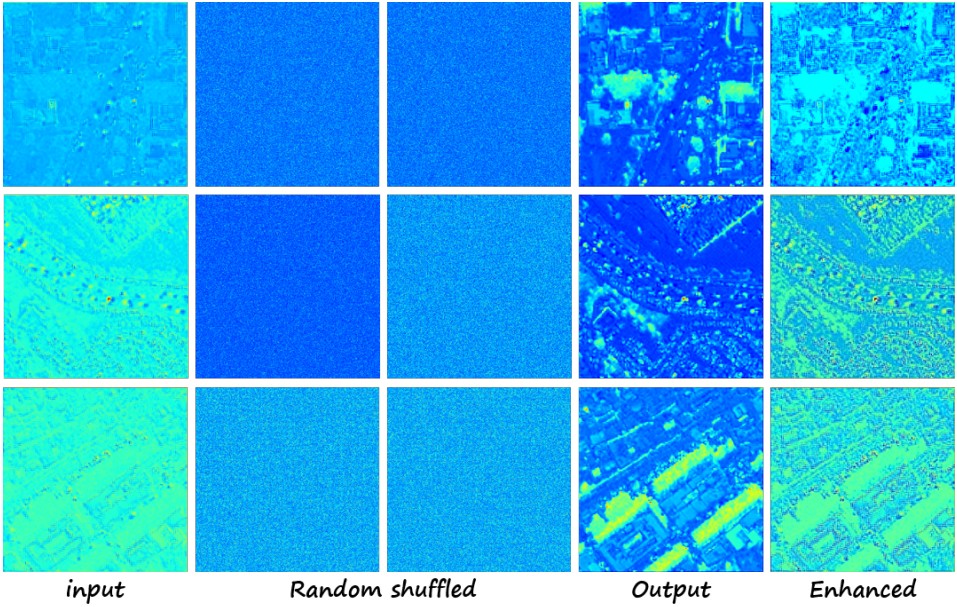

Figure 12: The feature visualization within RS-RWKV over pan-sharpening on the WorldView-III satellite.

