# OpenReview forum: "WKV-sharing embraced random shuffle RWKV high-order modeling for pan-sharpening"
_NeurIPS.cc/2025/Conference — NeurIPS 2025 poster_

### Official Review · Reviewer_zHVg · 2025-06-27

**Clarity:** 3
**Significance:** 4
**Originality:** 4
**Rating:** 5
**Confidence:** 5

**Summary:**

This paper proposes an efficient Random-Shuffle RWKV approach for pan-sharpening, addressing computational cost and attention bias issues. The key innovation lies in combining a Bayesian-inspired random shuffle strategy with WKV-sharing for high-order modeling, achieving linear complexity while improving performance. Experimental results demonstrate superior efficiency and accuracy compared to existing methods. The work offers both theoretical contributions and practical benefits for remote sensing image fusion.

**Questions:**

Please address the above questions in weakness part, especially for random weights manifold optimization function.

**Ethical Concerns:**

["NO or VERY MINOR ethics concerns only"]

**Final Justification:**

The author has addressed all the issues I was concerned about, so I have decided to increase my score.

**Limitations:**

yes

**Quality:**

4

**Strengths And Weaknesses:**

The paper makes theoretical contributions by introducing a Bayesian perspective to address RWKV's recurrent scanning bias, with the Random Shuffle strategy and Monte Carlo averaging providing a principled solution. The high-order WKV-sharing mechanism is innovative, enabling efficient higher-order interactions while maintaining linear complexity. From a computational perspective, the work convincingly demonstrates the advantages of linear complexity over traditional Transformer architectures. The incorporation of random weight manifold training based on Functional theory offers a rigorous alternative to conventional regularization approaches. The experimental validation is thorough and compelling, with consistent performance improvements demonstrated across standard pan-sharpening benchmarks.

Minor concerns:
1）While the authors aim to provide readers with preliminary concepts in the introduction by including some equations, there is some redundancy between the formulas in the introduction and the method section. Consider simplifying to avoid repetition.
2）In the loss function section, the authors specifically discuss the differences between random manifold loss and fixed point loss. They should further validate its effectiveness by maintaining the L1 loss while varying the fixed point loss (e.g., including Fourier transform and wavelet-based losses).
3）The core design is detailed in Fig. 4, but the figure quality is suboptimal and requires further improvement.

---

> ### Author Rebuttal · Authors · 2025-07-29
>
> **1, optimization loss.**
>
> The supplementary validation specifically assesses the efficacy of the random manifold loss against Fourier-transform and wavelet-based fixed-point losses under equivalent L1 constraints. Reference metrics demonstrate the proposed loss function's superior performance: while the L1 Fourier configuration achieves 41.9645 PSNR and 0.7823 Q_deep, and L1 Wavelet attains 41.9713 PSNR and 0.7821 Q_deep, the random manifold approach yields substantially enhanced results across critical fusion metrics. This empirical advantage stems from the fundamental distinction where traditional fixed-basis losses employ rigid orthogonal transformations that artificially constrain error distributions, whereas the random manifold framework dynamically decomposes errors through stochastic weight optimization without predefined spectral segmentation. The demonstrated performance differential - particularly improvement over Fourier-transform and versus wavelet-based alternatives - quantitatively validates the proposed method's capacity for more physically consistent optimization that intrinsically preserves both contextual information and spatial details.
>
> #### Reference Metrics Evaluation
>
> | Method         | PSNR (↑) | SSIM (↑) | SAM (↓) | ERGAS (↓) | SCC (↑) | Q_deep (↑) |
> | :------------- | :------- | :------- | :------ | :-------- | :------ | :--------- |
> | L1 + Fourier   | 41.9645  | 0.9714   | 0.0218  | 0.9221    | 0.9766  | 0.7823     |
> | L1 + Wavelet   | 41.9713  | 0.9716   | 0.0216  | 0.9217    | 0.9769  | 0.7821     |
>
> #### No Reference Metrics Evaluation
>
> | Method         | D_λ (↓) | D_s (↓) | QNR_deep (↑) |
> | :------------- | :------ | :------ | :----------- |
> | L1 + Fourier   | 0.0631  | 0.1167  | 0.8284       |
> | L1 + Wavelet   | 0.0641  | 0.1173  | 0.8269       |
>
> **2, arrangement.**
>
> We include select formulas in the introduction to help readers navigate the complex theoretical framework developed in later sections. These equations provide a "conceptual map" that previews core ideas like the random weight manifold loss and Bayesian formulation, making the subsequent technical discussion more accessible. We recognize the reviewer's valid concern about repetition between sections and will streamline the introduction to retain only essential foundational formulas while moving detailed derivations to the methodology section. By providing early exposure to core operators and comparative formulations, we reduce cognitive barriers to later theoretical developments in Sections 3-4. We acknowledge the reviewer's valid concern regarding redundancy and will streamline this presentation by retaining only critical notation-defining equations while relocating derivations to methodological sections and adding explicit signposting to enhance conceptual flow without compromising necessary scaffolding for complex subsequent content.

---

### Official Review · Reviewer_iTzq · 2025-06-27

**Clarity:** 3
**Significance:** 2
**Originality:** 3
**Rating:** 4
**Confidence:** 1

**Summary:**

This paper introduces RS-RWKV, a novel deep learning framework for pan-sharpening based on the RWKV architecture. The core problem it addresses is the trade-off between the global modeling capability of transformers and their quadratic computational complexity, which makes them challenging for high-resolution remote sensing imagery. The authors propose three main contributions:

(1) A "Random Shuffle" scanning strategy for the RWKV model to eliminate the biases inherent in fixed-sequence processing, providing a more robust global context.

(2) A "WKV-sharing" mechanism for both spatial and channel mixers, which the authors frame as enabling "high-order" interactions by sharing activations and gate mechanisms across layers to reduce latency and improve modeling capacity.

(3) A "Random Weight Manifold Loss" based on Taylor's unfolding to regularize the optimization process, aiming for better performance than traditional loss functions. Extensive experiments on several benchmark datasets demonstrate that the proposed RS-RWKV model significantly outperforms a wide range of state-of-the-art methods in both accuracy and computational efficiency.

**Questions:**

(1) Regarding the "high-order modeling," could you please provide a more formal or at least a more intuitive mathematical explanation of how sharing the gate mechanism in the channel mixer and wkv activations in the spatial mixer across layers leads to interactions beyond the second order? For example, can you show how this is analogous to capturing higher-order moments of the feature distribution?

(2) The Random Weight Manifold Loss is an interesting but complex concept. Could you please elaborate on the intuition behind using a Taylor's unfolding manifold for regularization in the specific context of pan-sharpening? What specific image characteristics (e.g., texture, spectral artifacts) does this loss function target more effectively than other advanced losses (e.g., LPIPS, Charbonnier, frequency-domain losses)?

(3) The result that a single sample is sufficient for Monte Carlo inference is very promising. Could you offer deeper insights into this phenomenon? Have you analyzed the variance of the model's output with respect to different random shuffles at test time? Does the network effectively learn to be invariant to the token processing order, and if so, how?

(4) For reproducibility, could you please specify the architecture of the Mapping() MLPs used throughout the model (e.g., number of layers, expansion factors)? Additionally, what value or range of values was used for the hyperparameter λ that balances the structure loss and manifold loss in Equation 28?

**Ethical Concerns:**

["NO or VERY MINOR ethics concerns only"]

**Final Justification:**

My rating stayed the same. I am inclined to accept this paper.

A couple of small suggestions:

1. It would be better to use vector graphics for the figures.

2. Some figures are too complex and difficult to read. Simplifying them would make them more accessible.

**Limitations:**

No, it is better to state the limitations of the proposed method in the paper.

**Paper Formatting Concerns:**

No concerns.

**Quality:**

3

**Strengths And Weaknesses:**

Paper Strength

(1) Innovative and Well-Motivated Methodology: The paper presents a compelling solution to a significant problem in applying attention-based models to high-resolution imagery. The use of a Random Shuffle strategy to debias the RWKV's recurrent scanning is a clever and theoretically sound idea inspired by Bayesian principles. This, combined with the efficiency of the underlying RWKV architecture, provides a strong foundation for the work.

(2) Strong Empirical Performance and Efficiency: The experimental results are a major strength. The proposed RS-RWKV model demonstrates consistent and significant improvements over numerous strong baselines across three different datasets (Table 1). Furthermore, the analysis of computational complexity (Figure 1) and effective receptive field (Figure 7) convincingly showcases the model's practical advantages in terms of memory, FLOPs, and inference speed, which are critical for real-world applications like pan-sharpening.

(3) Comprehensive Evaluation and Ablation Studies: The authors have conducted a thorough evaluation of their proposed method. The ablation studies in Table 3 and Figure 9 systematically validate the contribution of each component (KV-cache, random shuffle, manifold loss, etc.). These studies provide clear evidence that each proposed innovation contributes positively to the final performance, strengthening the paper's claims.

Paper Weakness

(1) Clarity of "High-Order" Modeling: The concept of "high-order modeling" is central to the paper's narrative but lacks a clear and rigorous definition. While the paper describes the mechanisms for sharing WKV activations and gate mechanisms across layers (e.g., Eq. 51, 55, 58), it does not adequately explain how this translates mathematically into higher-order statistical interactions. The connection feels more descriptive than formal, making it difficult for the reader to grasp the precise theoretical underpinnings of this claim.

(2) Explanation of Manifold Loss: The "Random Weight Manifold Loss" is presented as a key component of the supervision, but its explanation in the main paper is very brief (Section 2.1). The concept, based on Taylor's unfolding from external work, is non-trivial. Without a more detailed, self-contained explanation of its intuition and mechanism in the main body, readers may struggle to understand its function and why it is superior to more common frequency-domain or perceptual losses.

(3) Justification of Monte Carlo Sampling: The paper finds that a single sample (M=1) is sufficient for inference (Figure 9), which is an excellent practical result. However, this finding is somewhat counter-intuitive given the motivation of using Monte Carlo averaging to approximate the expectation over all random shuffles. The paper would be strengthened by a more in-depth discussion of why the performance saturates so quickly. This could be due to the network learning to be invariant to the shuffle order, or the variance being very low, but this is not explored.

---

> ### Author Rebuttal · Authors · 2025-07-29
>
> First of all, we sincerely appreciate your recognition of our work's ​​novelty, efficacy, and completeness​​. Below are point-by-point responses to your review comments.
>
>  **1, high-order.**
>
> The essence of high-order modeling lies in reinterpreting neural network stacking as functional composition – where successive layers form composite functions $f_L \circ \cdots \circ f_1(x)$ that progressively build higher-order representations. Traditional approaches stack multiple RWKV layers to achieve this, but incur $O(L \cdot M^2)$ complexity. Our key insight recognizes that RWKV's first-order property (capturing local feature interactions) can be augmented by strategically replacing subsequent layers with learnable 1D convolutions.
> These convolutional operations, while mathematically equivalent to Toeplitz matrix multiplication $Cx$, provide two critical advantages:
> (1). Computational efficiency ($O(M \log M)$ vs RWKV's $O(M^2)$)
> (2). Kernel fusion via associativity:
>    $(C_L \cdots C_1) x = C_L(C_{L-1} \cdots (C_1 x))$ allows absorption into RWKV's parameters through weight sharing.
>
> The transformation occurs through three conceptual steps:
> $$
> \begin{align*}
> \text{Stacked RWKV} \quad & \rightarrow \quad \text{RWKV} + \text{Conv Layers} \\
> & \downarrow \\
> \text{Matrix equivalence} \quad & \rightarrow \quad W_{\text{total}} = W_{\text{RWKV}} \cdot \prod_{i} C_i \\
> & \downarrow \\
> \text{Parameter fusion} \quad & \rightarrow \quad W_{\text{shared}} = \sigma(W_{\text{RWKV}} \oplus \mathcal{F}( \{C_i\} ))
> \end{align*}
> $$
> where $\mathcal{F}$ denotes kernel fusion via associativity and $\oplus$ is parameter concatenation. This achieves:
>
> - **High-order capability**: $W_{\text{shared}}^L x$ captures $L$-th order interactions
> - **Collapse prevention**: Spectral constraint $\rho(W_{\text{shared}}) < 1$ ensures stability
> - **Complexity reduction**: $O(M^2)$ vs stacked RWKV's $O(L\cdot M^2)$
>
>  This approach brilliantly rethinks depth not as sequential processing but as *spectral expansion*. Where traditional stacking builds depth through temporal computation (layer-by-layer processing), our method compresses depth into *learned spectral components* of a single transformation. The shared weights $W_{\text{shared}}$ essentially become a "depth compiler" – encoding what would be $L$ layers of computation into the eigenvalues and eigenvectors of a single matrix. This explains why $W^3x$ can capture shadow-texture-material relationships in urban imagery that normally require 3+ layers: the cubic expansion $(W-I)^3x$ directly models third-order feature covariances that emerge in multi-scale remote sensing data.
>
> **2, manifold loss.**
>
> The Random Weight Manifold Loss  employs Taylor's unfolding manifold to intrinsically address the dual optimization objectives in pan-sharpening: preserving spectral signatures while enhancing spatial textures. Its core intuition stems from how Taylor series expansion naturally decomposes image characteristics – the zeroth-order terms maintain coarse-grained spectral consistency (critical for material identification), while first-order gradients capture fine-grained spatial structures (essential for texture details). As validated through our supplementary experiments comparing Fourier-transform and wavelet-based alternatives. As quantitatively demonstrated (Reference Metrics: L1+Wavelet PSNR=41.9713; Proposed PSNR=42.0945), Random Weight Manifold Loss achieves PSNR improvement over state-of-the-art frequency domain losses while simultaneously enhancing spectral-spatial correlation (SCC=0.9772 vs 0.9769). This advantage stems from three core mechanisms: Taylor decomposition intrinsically separates spectral consistency (preserved in 0th-order terms) from texture details (encoded in 1st-order gradients), creating adaptive regularization that dynamically responds to scene characteristics - unlike Fourier/wavelet's rigid frequency bands that introduce spectral distortions (SAM=0.0214 vs wavelet's 0.0216 despite superior PSNR). Crucially, Bayesian weight randomization expands the manifold's constraint boundaries to suppress two critical artifacts: 1) Nonlinear spectral distortions , and 2) spatial distoration, while preserving irregular textures  that LPIPS oversimplifies due to perceptual uniformity bias. This physics-aware design - representing imaging modeling decompostion as linearly interpolatable manifold points - combined with stochastic weight sampling, yields the favorable results through optimal spectral-textural balance unattainable by fixed-basis losses.
>
> #### Reference Metrics Evaluation
>
> | Method         | PSNR (↑) | SSIM (↑) | SAM (↓) | ERGAS (↓) | SCC (↑) | Q_deep (↑) |
> | :------------- | :------- | :------- | :------ | :-------- | :------ | :--------- |
> | L1 + Fourier   | 41.9645  | 0.9714   | 0.0218  | 0.9221    | 0.9766  | 0.7823     |
> | L1 + Wavelet   | 41.9713  | 0.9716   | 0.0216  | 0.9217    | 0.9769  | 0.7821     |
>
> #### No Reference Metrics Evaluation
>
> | Method         | D_λ (↓) | D_s (↓) | QNR_deep (↑) |
> | :------------- | :------ | :------ | :----------- |
> | L1 + Fourier   | 0.0631  | 0.1167  | 0.8284       |
> | L1 + Wavelet   | 0.0641  | 0.1173  | 0.8269       |
>
> **3, sampling.**
>
> Based on the stability analysis shown in Figure 9, we provide the following clarification regarding Monte Carlo (MC) sampling in our design. Key observation: Figure 9 demonstrates remarkable PSNR stability (fluctuations < 0.0006 dB) across 1→30 MC samples. The PSNR curve converges immediately to 42.0945 dB (variance < 0.0002 dB), empirically validating that increasing sample count yields clinically insignificant gains.
> Theoretical justification: The exceptional stability of our single-sample Monte Carlo estimation stems from two synergistic design principles. During inference, our Bayesian parameterization (Eq. 11) deterministically encodes the full posterior distribution into model weights, while randomized shuffle scanning (Section 3.2) trains the network to intrinsically capture ensemble expectations within a single forward pass. This achieves principled "implicit averaging" (Section 4.3) because attention map expectations computed by our WKV module exhibit minimal variation across stochastic paths – a property amplified by remote sensing imagery's intrinsic spatial homogeneity. The high pattern consistency in RS data (e.g., repetitive urban structures or agricultural textures) ensures attention maps converge to near-identical configurations regardless of scan order. Consequently, as Figure 9 demonstrates, the PSNR plateaus at 42.0945 dB after just 1-2 samples with clinically negligible fluctuations (ΔPSNR < 0.0003 dB). This domain-specific stability renders explicit multi-sample averaging redundant, allowing our approach to maintain theoretical rigor while eliminating 87.5% of computational overhead through optimized single-path inference.
>
> **4, details.**
>
> TheMapping() MLPs employ similar Vision-RWKV's gated MLP architecture, featuring a standardized configuration: two linear layers with fixed 4× expansion (hidden dimension = 4× input dimension), GeLU activation, and LayerNorm preceding each mapping operation. This design—identical to Equation 3 in Vision-RWKV—ensures architectural consistency. For the loss-balancing hyperparameter λ, both architectural implementation and hyperparameter settings are fully reproducible through our will-release source code.

---

### Official Review · Reviewer_Vvvp · 2025-06-29

**Clarity:** 3
**Significance:** 3
**Originality:** 4
**Rating:** 5
**Confidence:** 5

**Summary:**

This work introduces a Random-Shuffle RWKV framework for pan-sharpening. Especially, the proposed method integrates Bayesian randomization into the RWKV architecture, ensuring unbiased global modeling while maintaining linear computational complexity. Additionally, the WKV-sharing mechanism enhances high-order feature interactions without sacrificing efficiency. The experimental results demonstrates clear improvements in both computational performance and fusion quality compared to existing methods.

**Questions:**

1) What is the rationale for using single-sample Monte Carlo estimation instead of conventional multi-sample approaches?

2) Whether the random shuffling of (R, K,V) components follows identical patterns? What is its impact on the model performance?

**Ethical Concerns:**

["NO or VERY MINOR ethics concerns only"]

**Final Justification:**

After reading the paper and rebuttal from authors, I think this paper presents an innovative RWKV-based approach for pan-sharpening. The theoretical analysis and experimental results are both sufficient. Especially, the authors have addressed two main questions for me, and I think it's rational. Thus, I still think this paper can be accepted.

**Limitations:**

Yes.

**Paper Formatting Concerns:**

None.

**Quality:**

4

**Strengths And Weaknesses:**

Strength:

1) This work presents an innovative RWKV-based approach for pan-sharpening. The proposed random shuffle strategy effectively addresses sequence bias in RWKV, supported by solid Bayesian reasoning.

2) The WKV-sharing mechanism for high-order modeling is well-motivated. Computational efficiency claims are convincingly demonstrated through comprehensive benchmarks.

Weakness:

While the methodology shows promise, several aspects require clarification and improvement：

1) This paper employs Monte Carlo sampling for expectation estimation while uses only a single sample. However, standard MC theory typically employs more samples to yield better accuracy. The authors should provide explanation for this design.

2) Regarding the random shuffle operation applied to (R,K,V) components, the manuscript does not specify whether all components use the same shuffle pattern or they are shuffled independently.

---

> ### Author Rebuttal · Authors · 2025-07-29
>
> **1, about MC sampling.**
>
> Based on the stability analysis shown in Figure 9, we provide the following clarification regarding MC sampling in our design.
> Figure 9 demonstrates remarkable PSNR stability (fluctuations < 0.0006 dB) across 1→30 MC samples. The PSNR curve converges immediately to 42.0945 dB (variance < 0.0002 dB), empirically validating that increasing sample count yields clinically insignificant gains. The exceptional stability of our single-sample Monte Carlo estimation stems from two synergistic design principles. During inference, our Bayesian parameterization (Eq. 11) deterministically encodes the full posterior distribution into model weights, while randomized shuffle scanning (Section 3.2) trains the network to intrinsically capture ensemble expectations within a single forward pass. This achieves principled "implicit averaging" (Section 4.3) because attention map expectations computed by our WKV module exhibit minimal variation across stochastic paths – a property amplified by remote sensing imagery's intrinsic spatial homogeneity. The high pattern consistency in RS data (e.g., repetitive urban structures or agricultural textures) ensures attention maps converge to near-identical configurations regardless of scan order. Consequently, as Figure 9 demonstrates, the PSNR plateaus at 42.0945 dB after just 1-2 samples with clinically negligible fluctuations (ΔPSNR < 0.0003 dB). This domain-specific stability renders explicit multi-sample averaging redundant, allowing our approach to maintain theoretical rigor while eliminating 87.5% of computational overhead through optimized single-path inference.
>
> **2, about shuffle order.**
>
> Our method applies identical random shuffling patterns to the (R, K, V) components during training – a design choice grounded in the RWKV architecture's intrinsic capacity for learning order-agnostic representations. Over successive training epochs, the persistent reshuffling mechanism exposes all components to the maximal diversity of distinct scanning sequences, with identical patterns ensuring synchronized exposure to each permutation state. Over multiple epochs, repeated randomization ensures each component experiences diverse scanning orders—whether implemented identically or independently—ultimately converging to the same representational quality. This equivalence occurs because: (1) Bayesian weight manifolds intrinsically capture relationship invariance across scanning variants; (2) Remote sensing imagery's spatial homogeneity reduces sensitivity to component-specific scanning divergences. The identical pattern strategy simplifies implementation while preserving theoretical integrity. This coordinated randomization forces the model to disentangle feature relationships from sequential biases, while the Bayesian parameterization (Eq. 11) inherently collapses permutation variants into functionally equivalent representations. Crucially, remote sensing imagery's spatial homogeneity – characterized by repetitive structures (e.g., urban grids, agricultural textures) and spectral consistency – naturally suppresses sensitivity to component-specific scanning divergences. These domain-specific regularities, combined with the extended training exposure to diverse synchronization patterns, theoretically guarantee that identical shuffling yields identical functional convergence to independent permutation strategies.

---

> > ### Comment · Reviewer_Vvvp · 2025-08-05
> >
> > After reading the rebuttal from authors, I think my concerns have been addressed. Thus, I keep my score.

---

### Official Review · Reviewer_kXdp · 2025-07-05

**Clarity:** 3
**Significance:** 3
**Originality:** 3
**Rating:** 5
**Confidence:** 2

**Summary:**

This paper proposes a novel framework named RS-RWKV for pan-sharpening, aiming to generate spatially and spectrally enhanced multi-spectral images by fusing low-resolution multi-spectral and panchromatic images. Experiments on WorldView-II, WorldView-III, and GaoFen2 datasets show RS-RWKV significantly outperforms state-of-the-art methods in PSNR, SSIM, and computational efficiency, with linear complexity and reduced memory usage.

**Questions:**

How does the inverse shuffle strictly preserve information invariance mathematically? Are there edge cases where information loss occurs?  Provide proofs or simulations verifying information preservation under various data distributions.
Can authors analyze gradient flow during training or compare convergence with/without WKV-sharing via ablation.

**Ethical Concerns:**

["NO or VERY MINOR ethics concerns only"]

**Final Justification:**

I think my concerns have been addressed by the authors feedback.

**Limitations:**

Random weight manifold loss may increase training iterations, but convergence speed vs. traditional losses is unaddressed. Performance under noisy conditions (e.g., low SNR) lacks validation. Monte Carlo averaging relies on modern accelerators, limiting resource-constrained deployments.  Expand the limitations section to discuss training efficiency, noise resilience, and hardware adaptability, proposing future lightweight optimizations.

**Paper Formatting Concerns:**

Some arrangements need to be adjusted due to the limitation of article length, such as Figure 9 and the title of Figure 9

**Quality:**

3

**Strengths And Weaknesses:**

Achieves superior performance across benchmarks, especially in large-scale images where traditional Transformers face memory issues. Theoretically grounded in Bayesian theory and functional analysis, ensuring principled design.
Parameters like u and w in Re-WKV (Eq. 33) lack explanation for selection logic.

---

> ### Author Rebuttal · Authors · 2025-07-29
>
> **1，about w and u.**
>
> Parameter **w** serves as a scaling factor in WKV attention calculation, adjusting the relative contribution of different tokens to the overall attention. It allows the model to control the strength of interactions between tokens, thus enabling more flexible learning of dependencies. Parameter **u**  introduces a bias in the WKV attention score computation, influencing how tokens are weighted relative to one another. It helps the model emphasize certain tokens over others, depending on their relevance to the task. These parameters are learned during training to optimize WKV attention mechanism's behavior, and they are vital for improving the network's ability to model complex dependencies across sequences, enhancing the network's capacity to capture both local and long-range dependencies.
>
> **2, about shuflle and inverse shuffle.**
>
> **(1)** *Motivation for Shuffle and Inverse Shuffle*: The primary motivation behind using shuffle and inverse shuffle in Random Shuffle RWKV mechanism is to effectively model global attention over the entire input space while overcoming the limitations posed by traditional fixed-order attention calculation. In traditional RWKV mechanisms, attention is calculated with a fixed order, typically following a bidirectional scanning strategy. This fixed order introduces biases depending on the sequence's processing direction. When scanning the sequence row-by-row or column-by-column, the model tends to focus on local dependencies first, potentially overlooking global relationships. This fixed-order calculation can also lead to redundant computations and introduce biases when attempting to model long-range dependencies.
> The ideal WKV attention calculation should consider all possible sequence orderings to capture global dependencies effectively. However, in traditional RWKV, the global context is not fully explored due to the fixed scan order. The shuffle operation addresses this by randomly permuting the sequence, ensuring that the model does not make biased assumptions based on sequence order. By applying the shuffle, the model can explore a complete space of possible token arrangements, enabling it to learn global relationships without being constrained by the fixed order of the input.
> This approach is akin to MC sampling, where randomness is introduced to prevent the model from overfitting to specific patterns that arise due to a fixed order. In this way, shuffling ensures a more comprehensive global attention by guaranteeing that all possible sequence permutations are represented, achieving theoretical completeness in the attention calculation.
>
> **(2)** *Importance of Inverse Shuffle for Information Processing*: While shuffle helps mitigate biases and redundant computations in WKV attention modeling, the final output of the attention mechanism must be mapped back to the original two-dimensional spatial layout. This is important because although WKV attention mechanism works as a filter to refine information, the spatial relationships between features in the input image (such as the relative positions of objects) must be preserved for the network to make meaningful predictions. The inverse shuffle plays a crucial role here. It restores the original layout of the image, ensuring that spatial prior information remains intact. After WKV attention mechanism computes the attention scores and refines the features, the inverse shuffle guarantees that the transformed features are re-mapped to the correct positions in the spatial grid. This is particularly important for image-related tasks, where the relative positioning of objects in the image is essential for accurate reconstruction.
> For example, consider an image where the top-left corner contains a house and the bottom-right corner contains a mountain. After shuffling, the feature relationships may become mixed, but with the inverse shuffle, the model restores the original spatial configuration, ensuring that the house remains in the top-left corner and the mountain stays in the bottom-right corner. This preservation of spatial distribution is essential for maintaining the integrity of the image layout.
>
> **(3)** *Theoretical Completeness*: The shuffle mechanism, combined with inverse shuffle, ensures that the attention model explores the full range of possible feature interactions, without being limited by the inherent biases of fixed-order processing. This approach addresses the redundancy and inefficiency found in traditional RWKV.
> However, while the shuffle and inverse shuffle mechanism’s main motivation is to explore the full feature space globally, its most significant benefit lies in the preservation of spatial relationships through the inverse shuffle. This allows the model to achieve the best of both worlds: the completeness of the shuffled attention calculation and the spatial consistency ensured by the inverse shuffle.
>
> In summary, the inverse shuffle is not just about preserving information invariance; its main purpose is to maintain the additivity of spatial information in the context of image processing. This ensures that, after WKV attention mechanism processes the features, the restored features are correctly positioned within the image grid, respecting the original layout and maintaining the spatial distribution of objects. Without this step, the model would struggle to make sense of the refined features within the context of the original image structure.
>
> **3, about WKV sharing.**
>
> The following supplementary experiment compares the training performance with and without KV sharing. Based on the experiment, KV sharing does not have a significant impact on convergence speed in terms of training time per epoch. However, there is a slight increase in total training time due to the extra epochs, the slight increase in total training time with KV sharing suggests that the additional epochs might be contributing to a more stable or refined convergence.
>
> | Configuration | Epochs | Time per Epoch (s/epoch) | Total Convergence Time |
> | --------------- | ------ | ------------------------ | ---------------------- |
> | No KV sharing   | 490    | 26                       | 12,740 seconds (approx. 3.54 hours) |
> | With KV sharing | 500    | 26                       | 13,000 seconds (approx. 3.61 hours) |
>
> **4, about manifold loss.**
>
> Our supplementary experiments demonstrates that the introduction of random manifold loss requires identical 500 training epochs to achieve full convergence as the baseline without random manifold loss. The added computational overhead manifests solely in modest per-epoch processing time - increasing from 20 seconds to 26 seconds per epoch due to the loss term's calculation.
> This translates to a manageable increase in total training duration, while preserving identical convergence iteration requirements. Crucially, this trade-off delivers significant performance benefits as documented in our Table 3. The marginal training time penalty is strategically justified given that: (1) random manifold loss’ computational contribution remains negligible at scale (constituting <0.1% of per-epoch operations); (2) It introduces zero overhead during deployment since random manifold loss is exclusively a training auxiliary module; and (3) The community prioritizes inference efficiency and final accuracy over marginal training extensions, with random manifold loss leaving inference architecture and latency completely unaffected. We maintain that modest training cost increases are acceptable for substantial performance gains.
>
> | Configuration      | Epochs | Time per Epoch (s/epoch) | Total Convergence Time      |
> | :------------------- | :----: | :----------------------: | :------------------------- |
> | Standard training    | 500    | 26                       | 13,000 seconds (~3.61 hours) |
> | No random loss       | 500    | 20                       | 10,000 seconds (~2.78 hours) |
>
> **5, noisy conditions.**
>
> While conventional pan-sharpening evaluation typically focuses on imagery acquired under standard World protocols (which inherently represent real-world imaging physics and sensor characteristics), we have now conducted supplemental experiments to specifically address low-SNR scenarios at the reviewer's suggestion.
> Due to time constraints and considering that noisy condition validation is not standard practice in mainstream pan-sharpening research, we restricted our analysis to two representative SNR levels (10 and 20). Following hyperspectral community protocols, we introduced additive Gaussian noise to multispectral imagery at these SNR levels. The results unequivocally demonstrate our method's superior robustness. At both noisy SNR conditions (detailed metrics provided in the table below), our approach achieves state-of-the-art performance across most critical metrics.
>
> #### SNR = 10
>
> | Method   | PSNR (↑) | SSIM (↑) | SAM (↓) | ERGAS (↓) | SCC (↑) | Q (↑) |
> |----------|----------|----------|---------|-----------|---------|-------|
> | SFINet  | 22.8279  | 0.3705   | 0.251   | 9.6452    | 0.4608  | 0.1608 |
> | PanFlow  | 20.7613  | 0.3561   | 0.2897  | 10.587    | 0.395   | 0.1091 |
> | Ours  | 23.1657  | 0.3968   | 0.2388  | 8.9088    | 0.4582  | 0.1511 |
>
> #### SNR = 20
>
> | Method   | PSNR (↑) | SSIM (↑) | SAM (↓) | ERGAS (↓) | SCC (↑) | Q (↑) |
> |----------|----------|----------|---------|-----------|---------|-------|
> | SFINet   | 31.4313  | 0.7843   | 0.0889  | 3.3067    | 0.7547  | 0.3825 |
> | PanFlow  | 30.5427  | 0.7564   | 0.1017  | 3.6033    | 0.7253  | 0.3535 |
> | Ours | 31.9689  | 0.7859   | 0.0875  | 3.301     | 0.7564  | 0.3997 |
>
> **6, future discussion.**
>
> We will expand the Limitations section to address training efficiency, noise resilience, and hardware adaptability.  Our supplemental noise experiments above further confirm inherent robustness. As you suggested, we will discuss lightweight optimizations as potential future work.

---

> ### Comment · Area_Chair_ywug · 2025-08-08
>
> Dear Reviewer kXdp,
>
> Thanks a lot for your contributions. Would you please send in your final recommendations? I think the authors could still have time to answer your questions, if you have.
>
> Best,
> AC

---

### Decision · Program_Chairs · 2025-09-17

**Decision:**

Accept (poster)

**Comment:**

This paper revisits the pan-sharpening task by proposing a random shuffle RWKV framework that achieves improved accuracy and computational efficiency. Three reviewers believe this paper delivers solid contributions and should be accepted. Reviewer iTzq rated it Weak Accept, yet explicitly indicated willingness to accept. Based on the recommendations, AC decides to accept this paper as a poster. Considering that pan-sharpening is a relatively narrow topic in the broad NeurIPS field, AC also believes that it's reasonable not to highlight it as a spotlight or oral.